# Insight-HXMT observations of jet-like corona in a black hole X-ray binary MAXI J1820+070

Bei You [1,2], Yuoli Tuo [3,4], Chengzhe Li[1], Wei Wang [1,2✉], Shuang-Nan Zhang [3,4✉], Shu Zhang[3], Mingyu Ge[3], Chong Luo[1,2], Bifang Liu[5,6], Weimin Yuan[5,6], Zigao Dai[7], Jifeng Liu[6,8,9], Erlin Qiao[5,6], Chichuan Jin[5,6], Zhu Liu[5,6], Bozena Czerny[10], Qingwen Wu[11], Qingcui Bu [3,12], Ce Cai[3,4], Xuelei Cao[3], Zhi Chang [3], Gang Chen[3], Li Chen[13], Tianxiang Chen[3], Yibao Chen[14], Yong Chen[3], Yupeng Chen[3], Wei Cui [15], Weiwei Cui[3], Jingkang Deng[14], Yongwei Dong [3], Yuanyuan Du[3], Minxue Fu[14], Guanhua Gao[3,4], He Gao[3,4], Min Gao[3], Yudong Gu[3], Ju Guan[3], Chengcheng Guo[3,4], Dawei Han[3], Yue Huang[3], Jia Huo[3], Shumei Jia [3], Luhua Jiang[3], Weichun Jiang[3], Jing Jin[3], Yongjie Jin[16], Lingda Kong[3,4], Bing Li [3], Chengkui Li [3], Gang Li[3], Maoshun Li[3], Tipei Li [3,4,15], Wei Li[3], Xian Li[3], Xiaobo Li[3], Xufang Li [3], Yanguo Li[3], Zhengwei Li[3], Xiaohua Liang[3], Jinyuan Liao[3], Congzhan Liu[3], Guoqing Liu[14], Hongwei Liu[3], Xiaojing Liu[3], Yinong Liu[16], Bo Lu[3], Fangjun Lu [3], Xuefeng Lu[3], Qi Luo [3,4], Tao Luo[3], Xiang Ma[3], Bin Meng[3], Yi Nang[3,4], Jianyin Nie[3], Ge Ou [17], Jinlu Qu[3], Na Sai [3,4], Rencheng Shang[14], Liming Song[3,4], Xinying Song[3], Liang Sun[3], Ying Tan[3], Lian Tao [3], Chen Wang [4,5], Guofeng Wang[3], Juan Wang[3], Lingjun Wang [3], Wenshuai Wang[17], Yusa Wang[3], Xiangyang Wen[3], Baiyang Wu[3,4], Bobing Wu [3], Mei Wu[3], Guangcheng Xiao[3,4], Shuo Xiao[3,4], Shaolin Xiong[3], Yupeng Xu [3,4], Jiawei Yang[3], Sheng Yang[3], Yanji Yang[3], Qibin Yi[3,18], Qianqing Yin[3], Yuan You[3,4], Aimei Zhang[3], Chengmo Zhang[3], Fan Zhang[3], Hongmei Zhang[18], Juan Zhang[3], Tong Zhang[3], Wanchang Zhang[3], Wei Zhang[4], Wenzhao Zhang[13], Yi Zhang[3], Yifei Zhang[3], Yongjie Zhang[3], Yue Zhang[3,4], Zhao Zhang[14], Ziliang Zhang[3], Haisheng Zhao[3], Xiaofan Zhao[3,4], Shijie Zheng[3], Dengke Zhou[3,4], Jianfeng Zhou[16], Yuxuan Zhu[3,19] & Yue Zhu[3]

A black hole X-ray binary produces hard X-ray radiation from its corona and disk when the accreting matter heats up. During an outburst, the disk and corona co-evolves with each other. However, such an evolution is still unclear in both its geometry and dynamics. Here we report the unusual decrease of the reflection fraction in MAXI J1820+070, which is the ratio of the coronal intensity illuminating the disk to the coronal intensity reaching the observer, as the corona is observed to contrast during the decay phase. We postulate a jet-like corona model, in which the corona can be understood as a standing shock where the material flowing through. In this dynamical scenario, the decrease of the reflection fraction is a signature of the corona's bulk velocity. Our findings suggest that as the corona is observed to get closer to the black hole, the coronal material might be outflowing faster.

---

A full list of author affiliations appears at the end of the paper.

During an outburst, a black hole (BH) X-ray binary usually displays transition between hard and soft states, according to the spectral properties of its radiation[1–4]. In the hard state, usually defined as the photon index $\Gamma < 2$ between 2 and 10 keV, the observed radiation primarily comes from the Comptonization by the hot electrons in the corona, which dominates over the weak, low-energy blackbody radiation from the disk. In the soft state with the photon index $\Gamma > 2$, the observed radiation then is characterized by a strong (disk) blackbody component below ~10 keV and a weak, high-energy tail of ~25% of the total bolometric luminosity[2,5–8]. MAXI J1820 + 070 (ASASSN-18ey) is a low-mass BH X-ray binary, newly discovered in X-rays with MAXI[9] on 11 March 2018[10]. In addition to X-ray, the source has also been observed in optical[11–15] and in radio[16,17]. Low-frequency quasi-periodic oscillations (LFQPOs) were found in both X-ray and optical bands[18–20]. The measurement of the radio parallax indicates that this source is located at a distance of 2.96 ± 0.33 kpc away from us[21]. Follow-up X-ray observations since its outburst were carried out by other X-ray telescopes, e.g., Swift[22], NuSTAR[23], and NICER[24]. The long-term and high cadence observation of MAXI J1820 + 070 by Hard X-ray Modulation Telescope (called Insight-HXMT)[25] was carried from 2018-03-14 (MJD 58191) to 2018-10-21 (MJD 58412)[26]. Figure 1 shows the Insight-HXMT HE (red)/ME (blue)/LE (green) light curves (Fig. 1a) and hardness intensity diagram (Fig. 1b) during the whole observations, displaying two outbursts. In the first outburst, MAXI J1820 + 070 rapidly rose from 2018-03-14 (MJD 58191) to 2018-03-23 (MJD 58200), and then gradually decayed until 2018-06-17 (MJD 58286). During this first burst, the source was in the hard state with the observed radiation being dominated by the high-energy photons (>20 keV; purple points in Fig. 1b). The second outburst started on 2018-06-19 (MJD 58288), with HE counts rising again to the peak on MJD 2018-07-01 (MJD 58300). After then, the source started to decay again until 2018-10-21 (MJD 58412). In this decay, the source mainly stayed in the soft state in which the LE photon rates dominated over ME/HE photon rates. The same Insight-HXMT observations have been used recently to study the timing properties of the outburst and LFQPOs were discovered above 200 keV, which was interpreted as due to the precession of a twisted compact X-ray jet[27].

Spectral and timing analysis of MAXI J1820 + 070 from MJD 58198 to MJD 58250 based on NICER reported that the profile of the broad Fe K emission line was almost unchanged, indicating the inner disk is very close to the BH (about 2 $R_g$, where $R_g = GM/c^2$ is the gravitational radius) and keeps constant[28].

Moreover, the thermal reverberation (soft) lags evolved to higher frequencies. Given these observational properties, it is suggested that it may be the corona that evolves during the outburst of this source, rather than the inner accretion disk, i.e., the corona might be contracting with time[28]. The detection energy band of NICER is 0.5–12 keV. However, the reflection component dominates over the X-ray spectra around 20–50 keV. The spectral fitting at this energy range could put crucial constraint on the reflection parameters[29], e.g., the reflection fraction and reflection strength, which can be used to probe the inner geometry of the accretion flow[30]. Moreover, the high-energy cutoff of the X-ray spectrum can be used to measure the electron temperature which is the important physical parameter to study the evolution of the accretion flow in the hard state. Insight-HXMT observes the broad-band X-rays from 1 to 250 keV, which enables us to probe the evolution of the accretion flow in detail by spectral and timing analysis. In this paper, we aim to reveal how the physical properties of accretion flow (corona + disk) evolve with time, mainly concentrating on the period when the radiation is dominated by high-energy photons most likely from the corona, by fitting the Insight-HXMT spectrum.

In this work, we show the evolution of the accretion flow in the hard state when the emission is dominated by high-energy photons from the corona, by analysing the observations of the first outburst of MAXI J1820 + 070 with Insight-HXMT. The derived reflection fraction $R_f$ is found to increase with time in the rising phase and decrease with time in the decay phase. In the scenario of a dynamical corona, this may suggest that as the corona contracts, i.e., the dissipation region gets closer to the BH, the coronal material might be outflowing faster.

## Results

We found that all spectra of MAXI J1820 + 070 in the first outburst can be approximately characterized by an asymptotic power law with a cutoff at high energy ≤100 keV. The broad continuum is thought to originate from the thermal Comptonization of the seed photons from the accretion disk by the hot plasma (hereafter corona) near the BH. Moreover, our preliminary spectral analysis to those spectra with the simple cut-off power law clearly reveals the line features at about 3–10 keV and the hump above 20 keV, with respect to the thermal Comptonization (see Fig. 2 and qualitative spectral analysis section in "Methods"). These features may be related to the illumination to the accretion disk. The Comptonized photons which illuminate the disk are reprocessed within the disk and then reflected off to

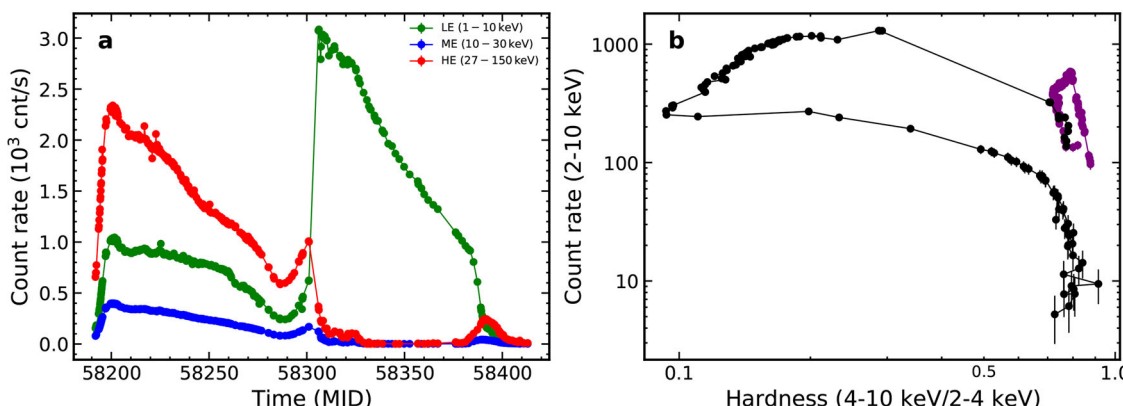

**Fig. 1 Insight-HXMT light curves and hardness-intensity diagram. a** Insight-HXMT light curves (in units of counts per second) of MAXI J1820 + 070 in HE (red, 20–250 keV), ME (blue, 5–30 keV) and LE (green, 1–15 keV) band. **b** The Insight-HXMT hardness-intensity diagram, defined as the total 2–10 keV count rate (in units of counts per second) versus the ratio of hard (4–10 keV) to soft (2–4 keV) count rates. The purple dots represent the first outburst from MJD 58192 to MJD 58286, which are fitted in this work.

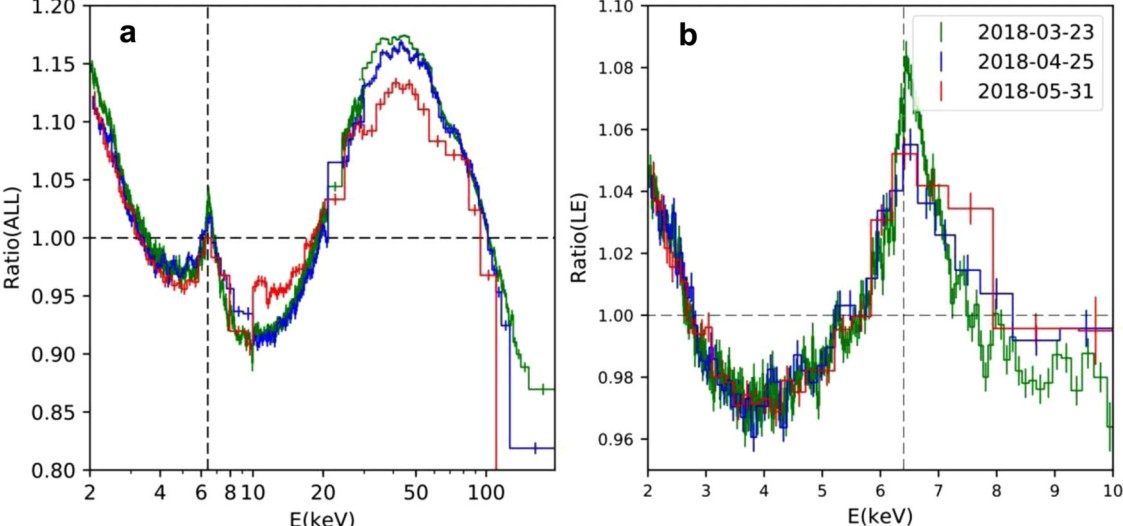

**Fig. 2 Ratio of the spectrum to the best-fitting cutoff power law. a** Ratio of the spectrum of three epochs to the best-fitting cutoff power law (cutoffpl in XSPEC) in 2–200 keV. Time runs from top to bottom, corresponding to the early, middle and late echo of this decay, i.e., 2018-03-23 (MJD = 58201, ObsID = P0114661003), 2018-4-25 (MJD = 58233, ObsID = P0114661028), 2018-05-31 (MJD = 58269, ObsID = P0114661060). **b** Ratio of the spectrum of the same epochs to the best-fitting power law in 3–10 keV. The vertical dashed line indicates the rest energy (6.4 keV) of the iron line. Fluorescence lines due to the photoelectric effect of electrons in the K-shell of silver are detected by the Si-PIN detectors of ME, which dominates the spectrum over 21–24 keV. Therefore, the spectrum over 21–24 keV is ignored.

the observer, resulting in the reflection with the characteristic iron line and hump features[31]. Therefore, we applied relxillCp[32] to account for the reflection model, which is the standard reflection model taking the relativistic effect into account. This reflection model contains both the direct emission from the corona and its reflection on the disk, which will be used in this work as the basic spectral model to fit the spectra of MAXI J1820 + 070 in the first outburst. The parameter settings of the relxillCp model are given in the configuration of the spectral fitting model section in methods.

relxillCp allows to fit the broad iron Kα emission line around 3–10 keV. However, in the residuals, we detect a clear narrow core to the iron emission (~6.4 keV), which was also found in NICER data[28] and NuSTAR data[33]. This narrow component may also originate from the reflection, but due to photoionization of neutral material or at least colder gas further from BH. So we additionally include another non-relativistic reflection model xillverCp to account for this narrow-line component. Moreover, we include diskbb[34] to account for the accretion disk radiation at low energy (~2–3 keV), and add tbabs[35] to accounts for the low galactic extinction[36] of $N_{\rm H} \sim 1.5 \times 10^{21}$ cm$^{-2}$. Therefore, the fitting modeling

$$\text{tbabs} * (\text{diskbb} + \text{relxillCp} + \text{xillverCp}) * \text{constant} \quad (1)$$

is applied in the spectral fits.

The spectra of 70 epochs, from 14 March 2018 (MJD 58191) to 17 June 2018 (MJD 58286), are fitted with the model described above. The best-fitting results are obtained by implementing a Markov Chain Monte-Carlo (MCMC) algorithm, and all spectra can be fitted well with the reduced $\chi^2 < 1.0$ (see spectral fitting method section in methods). In order to illustrate the spectral fitting, we plot one spectrum (MJD 58204, ObsID = P0114661006) with the best-fitting model in Supplementary Fig. 1, with the decomposition into the disk blackbody component, the Comptonization and reflection components from the reflection models. The evolution of the best-fitting parameters of interests for these good fits are plotted in Fig. 3. The electron temperature decreases from ~230 to 50 keV in the rise phase, and starts to slowly increase to ≤100 keV in the decay phase.

This tendency is in agreement with the results based on the combined spectra of MAXI/GSC and Swift/BAT[37,38].

Moreover, the best-fitting reflection fraction $R_f$ of the X-ray point source steeply increases up to about 0.5 in the rise phase and then slowly decreases to about 0.1 in the decay phase. In the reflection model, relxillCp, the reflection fraction is defined as the ratio of the coronal intensity that illuminates the disk to the coronal intensity that reaches the observer. The reflection fraction as well as some other parameters, e.g., ionization parameters, iron abundance, and the emissivity profile, have impacts on the shape of the reflection spectrum. This means that the increasing fraction of photons from the X-ray source illuminate the disk in the rise phase while the decreasing fraction of photons from the X-ray source illuminate the disk in the decay phase. In the hard state of GX 339-4, the reflection fraction is also found to be positively correlated with the source luminosity[30]. Assuming the X-ray source to be point-like static lamppost, the height of the X-ray point source can uniquely correspond to the reflection fraction[29], given the configuration of model parameters, i.e., BH spin, inner/outer radius of the disk. In this case, in the decay phase, the decrease in the reflection fraction corresponds to the increases of height.

However, the spatial extent of the corona which was estimated by the timing analysis of NICER data covering the early decay studied in this work, decreases or contracts with time[28]. It was previously discussed that the corona above the disk could be effectively ejected away from the accretion disk by the pressure of the reflected radiation, if the Comptonizing source is dominated by $e^{\pm}$ pairs[39]. This scenario has successfully explained both the weakness of the reflection with the reflection fraction $R_f < 1.0$ and hard X-ray spectrum with the photon index $\Gamma \sim 1.6$, in active galactic nuclei and X-ray binaries. When the corona, if being assumed as the point-like lamppost for simplicity, is accelerated outflowing away from the BH, the beaming effect will reduce the illuminating flux towards the accretion disk, which reduces the reflection fraction. In general, the lamppost geometry should be interpreted as a jet base, possibly a standing shock[40] which determines the corona position, but with material flowing through the standing shock. So the system should be

 3

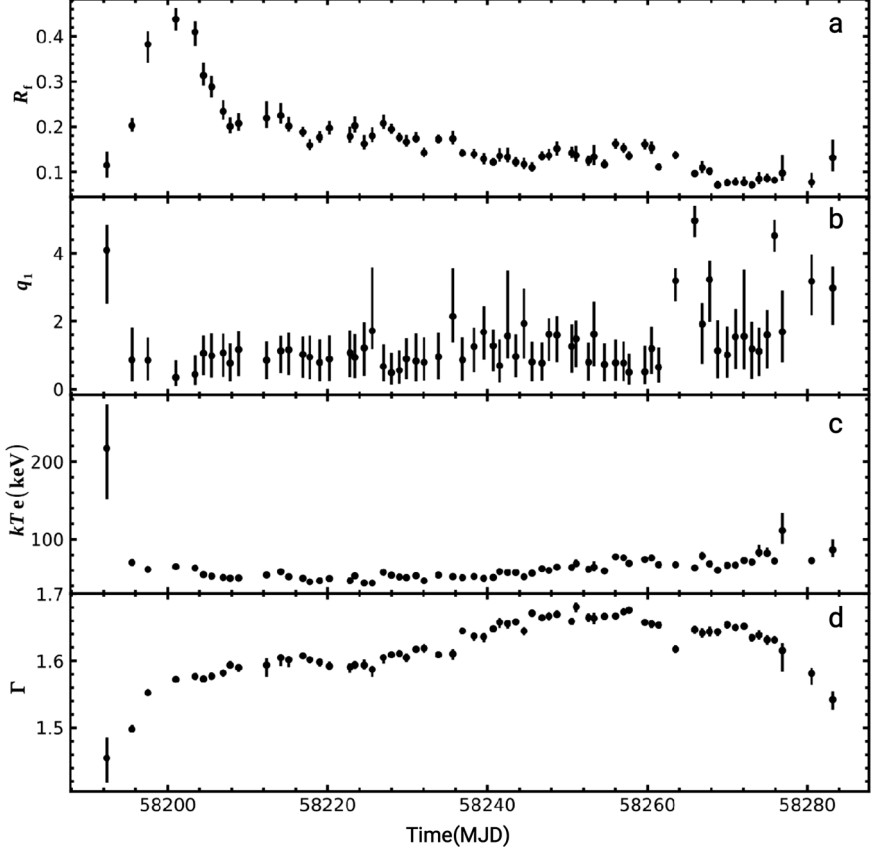

**Fig. 3 Time-evolutions of the free parameters in the best-fitting of the spectral model (1).** The evolution of the dimensionless reflection fraction $R_f$ which is defined as the ratio of intensity emitted towards the disk to that emitted to infinity, the dimensionless index $q_1$ of the emissivity profile, the electron temperature $T_e$ in units of keV, and the incident photon index $\Gamma$, are plotted in subpanels (**a**), (**b**), (**c**), and (**d**), respectively. Assuming an inclination angle $\theta = 63°$, the inner/outer radius of the reflection disk $R_{in} = R_{ISCO}$ ($R_{ISCO}$ is innermost stable circular orbit) and $R_{out} = 1000\,R_g$, the black hole spin $a = 0.998$. The black points correspond to the median of the values and the error bars correspond to 68% confidence interval, which is calculated using the corner[59] package to analyse the probability distributions derived from the MCMC chains. The uncertainties of the fitted parameters arise from both the statistical and systematic uncertainties.

characterized by two independent parameters: its position and bulk velocity. The corona position and its evolution with time could be constrained by the timing analysis[28]. In addition to that, the bulk velocity and its evolution with time can be constrained by the spectral analysis. In order to investigate the effect of the bulk motion of the corona on the reflection fraction, we calculate the reflection fraction by applying the package relxilllpionCp of relxill(v1.4.0), with the X-ray point source being located at height $H$ above a BH and outflowing with the velocity $\beta = v/c$. The light bending effect is taken into account. In Fig. 4, we plot the reflection fraction $R_f$ as a function of height $H$ and the bulk motion velocity $\beta$. It can be seen that, the reflection increases as the height decreases, as we expect in the normal lamppost model. In the mean time, for a given height, the reflection fraction gets smaller if the outflowing velocity increases. During the decaying phase of MAXI J1820 + 070 between MJD 58200 and 58286, X-ray timing analysis suggested the contracting of the corona with decreasing height[28], which should enhance the reflection fraction in the static lamppost model. However, the broad-band X-ray spectral results suggest that the reflection fraction decreases with time. Considering the effect of the bulk motion of the corona on the reflection fraction, the timing and spectral properties could be self-consistently explained in the scenario of the outflowing corona.

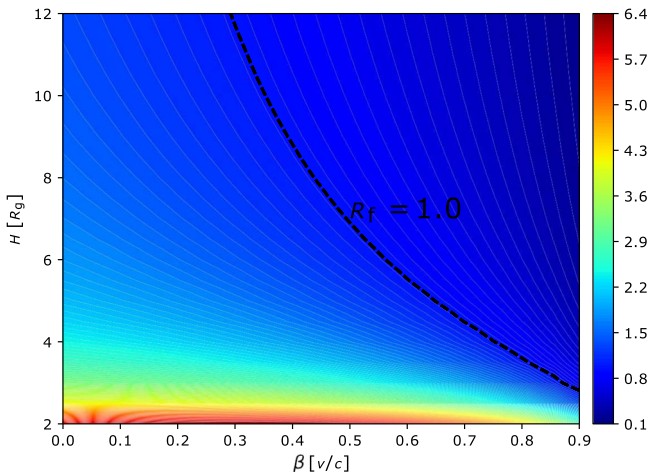

**Fig. 4 The dependence of the reflection fraction on the corona height and the bulk velocity.** The dimensionless reflection fraction $R_f$, is estimated as a function of the corona height $H$ (in units of $R_g$) and the bulk velocity $\beta = v/c$. The dashed line corresponds to the reflection fraction $R_f = 1.0$. The color bar represents the dimensionless reflection fraction, with the minimum and maximum being 0.1 and 6.4, respectively.

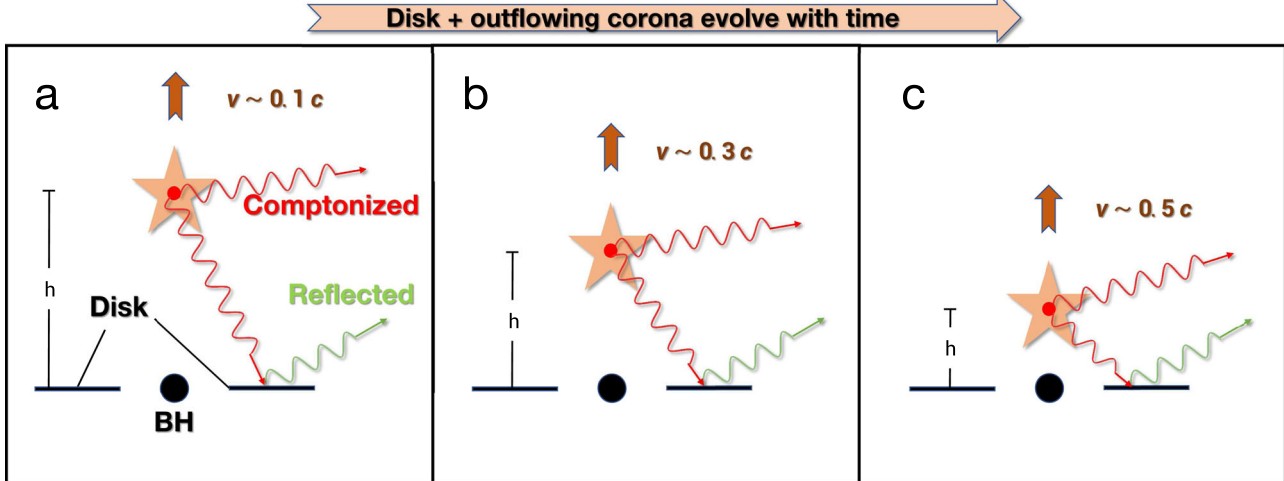

**Fig. 5 Schematic of the proposed jet-like corona in the decay phase.** 'BH' stands for Black Hole, and the symbol `star' represents the coronal region where the X-ray radiation comes from. The corona at height h can be understood as a standing shock where the material flowing through. The Comptonized hard photons (red arrow) from the corona illuminates the accretion disk, resulting in the observed reflection component (green arrow). Panel **a**, **b** and **c**, represents the peak, middle and end of the decay phase, respectively. As the corona contracts towards the black hole (e.g., from panel **a** to **b**) with decreasing height, the fitted reflection fraction decreases, which suggests that the bulk motion of the outflowing coronal material (sketched as an upward arrow) gets faster in the deeper gravitational potential well, with the increase of the outflowing velocity $\nu$ in units of light speed **c**. Note that the values of $\nu$ are not taken from the spectral fits, but for the sake of depicting faster outflowing corona.

When the corona contracts towards the BH, the decrease in the height will enhance the reflection fraction. However, the bulk motion of the corona could significantly reduce the reflection. If the bulk motion effect could dominate over the contracting effect, the contracting corona with the increasing speed of the outflow can explain both the shortening of the time-lag of the X-ray emission observed by NICER and the weakening of the reflection component observed by Insight-HXMT, although the mechanism of the contracting/acceleration of the corona is still unknown. For clarity, we also plot the schematic of the evolution of the corona with time in Fig. 5.

The corona in the lamppost geometry, should be interpreted as a jet base, possibly a standing shock[40], and was discussed in a number of previous studies of AGN and BH X-ray binary in the low/hard state[41]; this probably provides a physical interpretation to the compact X-ray jet revealed by the timing analysis of the same Insight-HXMT data used here[27]. The evolution of the outflowing corona should be studied in terms of both its position and bulk velocity, which can help us to better understand the acceleration/deceleration of the particles near a BH. The X-ray temporal/timing analysis can provide us the spatial information of the relative geometry of the corona. The broad band spectral fittings which derived variations of the reflection fraction, help to probe the dynamical properties of the corona, i.e., if the corona is outflowing/inflowing. Therefore, this work discovered that the corona outflows faster as it contracts towards to BH, suggesting that the hard X-ray emission region in the jet base or standing shock is formed at a closer distance to the BH for faster outflow. This scenario could be applied to other BH X-ray binaries and AGNs in which the coronas are active, at least to diagnose if the outflowing/inflowing corona is accelerating or decelerating.

## Discussion
**On the evolution of the spectral parameters.** In the relxillCp model, the emissivity profile is assumed to be broken power law of two indexes $q_1$ and $q_2$ with the break radius $R_{\rm br}$. In our spectral fits, for simplicity, we assume the index $q_2 = 3$ for the region where $R > R_{\rm br}$. The emissivity profile is roughly flattened with the

index $q_1 < 1.0$ within the broken radius $R_{\rm br} \sim 20–60 R_{\rm ISCO}$ (where $R_{\rm ISCO} \sim 1.23 R_{\rm g}$ for $a = 0.998$). Given the predictions by ref.[42] (e.g., their Fig. 10 and 12), the flattened emissivity profiles suggest that the X-ray corona source might be spatially extended with the size of tens of $R_{\rm g}$ in radius. The emissivity profile for a point source is predicted to be much steeper with the index $q \sim 6–8$ in the inner region (e.g., $R < 3 R_{\rm g}$) before the profile flattens off (where $R < R_{\rm br}$). Such a twice-broken power-law profile was observed in the narrow-line Seyfert 1 galaxy 1H 0707-495 by fitting the relativistically broadened emission lines from the disk[43]. The derived small index with $q_1 < 1$ in the inner disk region from our spectral fits may be due to: (1) relxillCp uses a single broken-power law, so the index $q_1$ is the emissivity-weighted value for the region $R < R_{\rm br}$; (2) the index $q_1$ and the inner radius are somewhat degenerated with each other. On one hand, if the emissivity profile falls steeply with the index $q_1 \sim 6 - 8$, then in order to reproduce the flux levels emitted from the innermost regions, the disk must be truncated at a larger radius. On the other hand, if the emissivity profile is flat with a smaller index $q_1 < 1$, then the disk can extend to the innermost regions with a smaller truncation radius, to match the flux levels[42]. In our spectral fits, the inner radius is assumed to be small with $R_{\rm in} = R_{\rm ISCO}$, for simplicity. If we assume $R_{\rm in} = 20 R_{\rm g}$ instead and refit the spectra, the emissivity profiles then are required to be steep with the index $q_1 \sim 6–8$.

The reflection fraction in relxillCp is smaller than unity, which requires the spatially extended corona to be relativistically outflowing to overcome the increasingly important light-bending effect, since the corona is suggested to be contracting from NICER lag-observations[28]. Since the reflection fraction decreases along with the decay, the corona may be outflowing faster as it contracts over the time. The time evolution of the ionization parameter $\log \xi$, the abundance $A_{\rm Fe}$ with respect to the solar value, the constant for ME and the constant for HE are plotted in Supplementary Fig. 2. For these parameters, the correlations with respect to each other are also investigated with the Spearman's rank test, which are plotted in Supplementary Fig. 3. The probabilities of null correlation between the reflection fraction and the other three parameters ($A_{\rm Fe}$, $\rm Const_{ME}$, $\rm Const_{HE}$)

are $7.5 \times 10^{-21}$, 0.02, and 0.02, respectively. The two constant factors of ME/HE do not show obvious evolution over the epochs, and the reflection fraction is uncorrelated with the constants of ME/HE. Together with the fact that there is no degeneracy between the reflection fraction and these factors, we conclude that the trend of the reflection fraction reported in this work is unlikely associated with the constant factors of ME/HE.

The best-fitting values of the iron abundance are high with $A_{Fe} \geq 5$, and the fits prefer changing in iron abundance which roughly increases along with the decay, resulting in an apparent correction with the reflection fraction with the p-value of $7.5 \times 10^{-21}$. Such supersolar abundances also appear in the spectral fitting of other BH X-ray binaries, e.g., GX 339-4[44] and Cyg X-1[45]. It was suggested that the derived supersolar abundances are not physical, but may be caused by the limitation of the assumed density in the reflection model[46,47]. Both the density and the iron abundance of the accretion disk have impacts on the emergent reflection spectrum. As the density increases, the rise in free-free absorption leads to an increase in temperature, causing extra thermal emission at soft X-ray energy $E < 10$ keV[46]. This effect could become significant when the density is high ($n_e > 10^{19}$ cm$^{-3}$) (see Fig. 7 in ref. [47]), although it was emphasized that the microphysics in the current reflection models is only known to be accurate up to $n_e = 10^{19}$ cm$^{-3}$ (see ref. [46] for more details). The effect of the abundance on the reflection spectrum is studied, using the relxillCp model. The increase in the iron abundance from $A_{Fe} = 1$ to $A_{Fe} = 10$, will also cause the increase in the soft X-ray emission of the reflection for the highly ionized disk in the hard state (see Supplementary Fig. 4a), mimicking the effect of the disk density increase. Therefore, the soft X-ray emission could be reproduced by either high density or high abundance in the accretion disk[46,47]. It was shown that the high-density model with the solar value of iron abundance, or the low-density model with supersolar abundance, could both fit the NuSTAR spectra of Cygnus X-1[47]. In the current reflection model relxillCp which is used in this work, the constant density is fixed at constant value $n_e = 10^{15}$ cm$^{-3}$, which is much lower than the typical values in the standard thin disk model[48], $n_e \geq 10^{19}$ cm$^{-3}$. Along with the decay of the outburst studied in this work, the radiation pressure in the inner disk should decrease and thus the gas density should increase[46,49]. Therefore, the spectral fits with the much lower and constant density in the reflection model should lead to artificially supersolar and also increasing iron abundance along with the outburst decay, consistent with the spectral fitting results in this work. Deriving the physical values and evolution of the iron abundance and disk density with the spectral fitting, however, requires new atomic data in high-density to be implemented in the reflection model, which is beyond the scope of this work. Nonetheless, in order to demonstrate that the observed trend of the reflection fraction is physical and independent of the expected artificial correlation between the reflection fraction and the iron abundance, we tried the case of the constant iron abundance, e.g., $A_{Fe} = 5.0$. In this case, the reflection fraction still decreases along with the outburst decay (see Supplementary Fig. 4b), although fixing $A_{Fe}$ is not supported by F-test, due to the problems we discussed above.

The best-fitting values of the ionization parameter are high with $\log \xi \geq 3.8$. The ionization parameter is defined as the ratio of the illuminating flux and the electron number density ($n_e = 10^{15}$ cm$^{-3}$). According to this definition, changing the ionization parameter will not only redistribute the reflected photons over energy (i.e., reshaping the X-ray spectroscopy), but also the number of the reflected photons (i.e., increasing/decreasing the spectrum flux). Therefore, in the current relxill model, the reflection spectrum is normalized to the constant illuminating flux $F_0 = \xi n_e/4\pi$ where $\xi = 1$ erg cm s$^{-1}$, and the illuminating flux (which takes the reflection fraction into account) from the nthcomp is normalized to this constant illuminating flux.

Then, the total spectrum can be simply calculated by combining the reflection spectrum with the nthcomp spectrum. After normalizing the spectrum in the relxill model, the ionization parameter is not proportional to the illuminating flux[29,32]. In photoionization modeling of the relxill model, it is assumed that $\log \xi$ completely characterizes the X-ray spectroscopy, regardless of the actual values of density or flux[46]. Therefore, the highly ionized disk in Supplementary Fig. 2a is not in conflict with the weak illuminating flux, although the faster outflowing corona provides a weaker illuminating flux. A highly ionized disk for low flux at 3–78 keV was also found in the NuSTAR spectrum of MAXI J1820 + 070[33]. Instead, the large values of the ionization parameter is possibly caused by the over-estimated iron abundance as discussed above. The 0.01–10 keV flux would be reduced due to the increase of the continuum opacity, as the iron abundance increases to account for the deficiency of the constant density in the reflection model[46]. In order to compensate for this effect in the spectral fits, the degree of ionization in the disk would increase. This is because in the case of high ionization, the illumination continuum will not be highly absorbed by the photoelectric opacity, which leads to the relatively strong reflection continuum in 0.01–10 keV.

The broad component of the iron line visually appears stable throughout the decay, while the strength of the narrow core decreases with time, which is consistent with the observations of NICER and NuSTAR[28,33]. The broadening/narrowing of the broad iron line can also be quantitatively evaluated with the equivalent width (EW), a measure of the relative strength of the line profile. The formula of calculating the EW of the iron line (including narrow component) is as follows:

$$EW = \int_{E_{min}}^{E_{max}} \frac{F(E) - F_c(E)}{F_c(E)} dE, \qquad (2)$$

where $F(E)$ is the total flux and $F_c(E)$ is the total flux of the continuum under the line. $E_{max}$ and $E_{min}$ are the upper and lower energy band limits of the integration, respectively. $F(E)$ is taken from the best-fitting model spectrum, and the continuum $F_c(E)$ is approximated by a power law connecting $E_{max}$ and $E_{min}$[32]. Here, we take $E_{min} = 4$ keV and $E_{max} = 10$ keV, with the assumption that the iron line takes place in this region. Note that the simplification of the powerlaw continuum and the particular choice of integration limits are somewhat arbitrary, mainly considering the decomposition of the best-fitting spectrum (see Supplementary Fig. 1) and the ratio plot (see Fig. 2). In Supplementary Fig. 5a, we plot the time-evolution of the iron line EW from the best-fitting spectral model (relxillCp + xillverCp). It turns out that the iron line EW is roughly stable with ~200 eV during the first half of the decay (covering the epochs of NICER in ref. [28]), while there is some scatter.

Furthermore, as we discussed above, the X-ray spectroscopy (including the relative strength of the iron line) by relxillCp does not depend on the actual illuminating flux, EW then is insensitive to the reflection fraction. As discussed above, the broad Fe K$\alpha$ line appears stable throughout the decay, while the relative strength of the narrow core decreases with time. In order to understand the evolution of the X-ray spectroscopy of Fig. 2 in terms of the reflection models, in Supplementary Fig. 5b and c, we plot the time-evolution of EW from the best-fitting model spectrum of relxillCp, and the normalization of xillverCp. It shows that the constant broad iron line can be maintained by the evolution of relxillCp. The weakening of the narrow line as well as that decrease in the amplitude of the hump are attributed to the decrease in the normalization of xillverCp.

Supplementary Fig. 6 plots the time evolution of the diskbb parameters, i.e., the temperature at inner disk radius $T_{in}$ in units

of keV, the normalization and the diskbb flux in units of erg/cm$^2$/s. The disk inner temperature increases from 0.4 to 0.6 keV during the decay, although there is significant degeneracy between the inner temperature and the normalization of diskbb. We note that, the seed photon temperature $kT_{bb}$ in relxillCp is fixed at 0.05 keV, which is much lower than the best-fitting values of $T_{in}$. In order to study the effect of $kT_{bb}$ on the results, we freeze $kT_{bb} = 0.5$ keV. It turns out that the difference in $kT_{bb}$ does not have a strong effect on other parameters. We use cflux in XSPEC to estimate the diskbb flux in 0.01–100.0 keV and its time-evolution. We found that the disk flux increases with time until around MJD = 58,250, and then evidently decreases until the end of the decay. In our spectral model, the inner radius of the accretion disk is assumed to be constant at ISCO. We note that the inner radius can be derived from the diskbb parameters, e.g., GX 339-4[30] and XTE J1550-564[50]. In ref. [50], the inner radius was estimated with the photon flux of the direct disk component and the Comptonized disk component, taking into account the Comptonized disk photons. In this work, we also estimate the disk inner radius from the spectral fits. Although the observed Comptonization and the reflection flux can be fitted with the reflection model, it is hard to determine the Comptonized disk flux, which depends on the geometry of the corona with respect to the disk. Therefore, we introduce a constant factor $\lambda$ which is multiplied by the direct disk flux $F_d$ to estimate the Comptonized disk flux $F_c$, i.e., $F_c = \lambda F_d$. Then, the inner radius can be calculated as follows (see the Appendix in ref. [50],

$$F_d + F_c = (1 + \lambda)F_d = 0.0165 \times \frac{r_{in}^2 \cos i}{(D/10\ \mathrm{kpc})^2} \times \left(\frac{T_{in}}{1\ \mathrm{keV}}\right)^3 \text{ photons s}^{-1}\text{ cm}^{-2},$$

(3)

where the source distance $D = 2.96$ kpc, $i = 63°$ are adopted from the observational measurements[21], and the direct disk flux in 0.01–100 keV (in units of photons s$^{-1}$ cm$^{-2}$) from diskbb is obtained with the flux command in XSPEC. In Supplementary Figure 6d, we plot the derived $R_{in}$ (in units of $R_g$, assuming BH mass $M = 10 M_\odot$) along with time, for the case of $\lambda = 0$. It can be seen that, if only considering the direct disk flux, the inner radius of the disk is indeed very close to ISCO of BH. And if the Comptonized disk flux is comparable or dominates over the direct disk flux, i.e., $\lambda = 1$ or 10, respectively, $R_{in}$ should be enlarged by a factor of ~1.4 or 3.3. In this case, the derived inner radius may marginally contradict the assumption of an untruncated disk. Note that constraining the normalization of diskbb is tricky because of strong coupling with the inner disc temperature and the absorption column, and the estimate of the inner radius from diskbb also depends on the boundary condition and the color hardening factor. These may contribute to the uncertainty on the estimate of the inner radius. In ref. [33], the inner radius is one of the free parameters affecting the shape of the reflection spectrum (not from the disk parameter), which is fitted to the NuSTAR spectra of MAXI J1820 + 070.

During the decay of MAXI J1820 + 070, the Comptonization component is softened with the increase of the photon index, as the reflection fraction decreases with time. Considering the primary X-ray source as the blob of the plasma, the photon index of the Comptonized radiation emerging from the blob is determined by the Compton amplification factor "$A$" (which is defined as the ratio between the blob luminosity and the disk seed luminosity intercepted by the blob) taken in the plasma's comoving frame[51]. Moreover, it was shown in ref. [39] that this amplification factor depends on not only the blob outflowing velocity $\beta$ (as we discussed in this work) but also the geometrical factor $\mu_s$ of the blob (see their Fig. 1). The geometrical factor $\mu_s$ describes the relative geometry of the blob with respect to the

disk, with $\mu_s = 0$ corresponding to the case of a slab geometry while $\mu_s = 0.5$ roughly corresponding to a blob with radius of order its height. In other words, the decrease of $\mu_s$ corresponds to the increasingly strengthened disk flux intercepted by the blob. In the decay of MAXI J1820 + 070, the corona contracts with time[28], i.e., the corona getting closer to the disk, strengthening the intercepted disk flux by the corona with the decrease of the geometrical factor $\mu_s$. Therefore, on one hand, the Compton amplification factor can decrease due to the decrease of $\mu_s$, leading to the increase of the photon index (see Eq. 10 in ref. [39]); on the other hand, the Compton amplification factor can increase due to the increase of the outflowing velocity $\beta$, leading to the decrease of the photon index (see Supplementary Fig. 7b). In the decay of MAXI J1820 + 070, if the effect of the geometrical factor dominates over that of the outflowing velocity, the resultant photon index will increase, as found in our spectral fitting.

**On the effect of system parameters.** In our spectral fits, the BH spin is fixed at $a = 0.998$. However, it is also possible that the BH spin is low, if the accretion disk extends to the innermost stable circular orbit (ISCO), i.e., $R_{in} = R_{ISCO}$, provided that the inner radius $R_{ISCO}$ is fitted to be around 5 $R_g$ over the decay[33]. Spectral fits to NICER data of MAXI J1820 + 070 in the soft state suggests a low spin $a < 0.5$ (ref. [52]). In order to study the effect of the low spin on the evolution of the corona, we refit six observations that cover the decay of this outburst with the same model, but fixing the spin $a = 0.5$. It turns that the evolution of the corona remains (see Supplementary Fig. 8a), i.e., the reflection fraction is also smaller than unity, and decreases with the decay, which requires the relativistic outflowing motion of the corona.

The inclination is fixed to the constant value $\theta = 63°$ based on the jet/optical measurements. However, the inner disc/jet/orbital plane does not necessarily align. In order to study the constraint of spectral fits on the inclination and its effect on the evolution of the corona, we refit six observations that cover the decay of this outburst with the same model, but allowing the inclination to be free. It turns that, the spectral fits with the standard relativistic reflection model relxillCp also prefer to large values of the inclination, and the evolution of the corona remains (see Supplementary Fig. 8b and c), i.e., the reflection fraction is also small than unity and decreases during the decay phase.

**Justification of the outflowing corona.** Recently, the outflowing velocity of the lamppost X-ray primary source is taken into account in the package relxilllpionCp of relxill(v1.4.0). In principle, we could refit the Insight-HXMT data to directly infer the outflowing velocity and its evolution over time. We found that the outflowing velocity and the height cannot be uniquely determined in the spectral fits. Therefore, deriving the evolution of the outflowing velocity may require that the height of the corona can be estimated from other measurements. e.g., timing analysis. However, this is difficult to achieve due to the following two facts: (1) the effective area of LE detector in Insight-HXMT is one order of magnitude smaller than that of NICER, so that the derived time-lags in low-energy band is not precise enough; (2) converting the lag into a light travel time distance between the corona and the accretion disk is not straightforward. A number of effects, e.g., the geometry of the system, the viewing angle to this source and relativistic Shapiro delay, etc, have to be well studied before the reliable measurement of the light travel time can be derived[28]. Nonetheless, we could fix the height of the lamppost X-ray source at constant value, e.g., $H = 7 R_g$, and refit the spectra during the decay to directly infer the outflowing velocity and its evolution over time. It is shown in Supplementary Fig. 7a that the

outflowing velocity indeed increases from ~0.0 $c$ to ~0.8 $c$ along with time, resulting in the decrease of the reflection fraction.

Moreover, we simulate the energy spectra for NuSTAR observation, as a function of the lamppost height and outflowing velocity, using the reflection model relxilllpionCp which takes the outflowing of the corona into account. The EW of these simulated NuSTAR spectra is then estimated, which turns out to be fairly constant (see Supplementary Fig. 9). Given the simulation results in Supplementary Figs. 4a and 9, it can be seen that the EW of the iron line depends on not only the height and the outflowing velocity of the lamppost but also the iron abundance and ionization of the reflection disk[53].

**Comparison to NuSTAR spectral analysis.** NuSTAR spectra of MAXI J1820 + 070 in the decay were reported and analysed in ref. [33]. The two lamppost point sources (relxilllpCp) with different heights, as the reflection model, were used in their spectral fits. It was shown that, during the decay, the height of the lower point source remains constant at about ~$4R_g$, while the height of the upper point source decreases from 100 $R_g$ to a few $R_g$. We also used the two lampposts model to fit Insight-HXMT spectra, and found that the derived height of the upper lamppost decreases with the decay as well, which is consistent with the results of NuSTAR. Note that, in relxilllp(Cp) and relxilllpion(Cp), the disk outer radius cannot exceed 1000 $R_g$. Therefore, when the corona is high, e.g., $H > 100R_g$, a fraction of photons will not hit the disk, which results in the reflection fraction being lower than unity. This should be distinguished from the case of the corona being relativistically outflowing at low height $H < 100R_g$. In the latter case, the reflection fraction will also be small with $R_f < 1$, but now it is due to the beaming effect of the relativistically outflowing motion.

However, in the lamppost model relxilllpCp, the corona is assumed to be stationary. The conversion from height to reflection fraction is not self-consistent between the stationary corona model (relxilllpCp) and the moving corona model (relxilllpionCp), since the corona discussed in this work might be moving at different heights, along with the evolution of the outburst. Taking the spectrum of the later epoch MJD = 58271 (obsID = P0114661061) as an example, when the corona was believed to be contracting closer to BH[28], we found that, both the stationary corona with the height $H \sim 3R_g$ using relxilllpCp, and the outflowing corona at different heights (e.g., $H \sim 17R_g$, $v \sim 0.5c$; $H \sim 45R_g$, $v \sim 0.4c$) using relxilllpionCp, can provide the equivalently good fits to the data. In the former case, the reflection fraction is high $R_f = 3.3$, while in the latter case, the reflection fraction is low $R_f \sim 0.6$. These comparison studies above confirm that the decrease of the height alone cannot explain the observed reflection evolution, which also requires the outflowing corona reported in this work.

Meanwhile, in the case of an outflowing corona (relxilllpionCp), the height and the outflowing velocity cannot be uniquely determined from the spectral fits, as discussed above. Nonetheless, at a later epoch of the outburst (e.g., MJD = 58271), the corresponding reflection fractions are roughly the same $R_f \sim 0.6$, and the fitted velocity is moderate with $v \sim 0.5c$. In contrast, at the peak of the outburst, e.g., MJD = 58204 (obsID = P0114661006), the fitted reflection fraction is $R_f = 1.2$, and the corona prefers to be stationary, using relxilllpionCp model. Therefore, although it is not easy to uniquely determine the height and the outflowing velocity, the resultant reflection fraction decreases during the decay, which is consistent with our results in the spectral fits of Insight-HXMT using relxillCp.

Here, we also directly fit the reflection fraction from the spectral fits to NuSTAR data with the relxillCp + xillverCp model, as is done for Insight-HXMT in this work. The inclination

angle is fixed at constant large value $\theta = 63°$. The two diskbb components are used to account for the differences between FPMs due to a thermal blanket tear in FPMA[54]. It turns out that the the reflection model which consists of the relativistic reflection relxillCp and non-relativistic reflection xillverCp can also fit the NuSTAR spectra well with reduced $\chi^2 \leq 1.1$, and the best-fitting values of the parameters are given in Supplementary Tables 1, 2 and 3. The index of emissivity profile $q_1$ are pegged at zero. This also suggests that the corona is spatially extended, as seen in Insight-HXMT fits. More importantly, the reflection fraction turns out to decrease with the decay from 0.8 to 0.3 (see Supplementary Fig. 10), which has the same trends as the Insight-HXMT results in this work (see Fig. 3).

## Methods

**Data reduction.** Insight-HXMT consists of three groups of instruments: High-Energy X-ray Telescope (HE, 20–250 keV), Medium Energy X-ray Telescope (ME, 5–30 keV), and Low Energy X-ray Telescope (LE, 1-15 keV). HE contains 18 cylindrical NaI(Tl)/CsI(Na) phoswich detectors with a total detection area of 5000 cm$^2$; ME is composed of 1728 Si-PIN detectors with a total detection area of 952 cm$^2$; and LE uses Swept Charge Device (SCD) with a total detection area of 384 cm$^2$. There are three types of Field of View (FoV): 1° × 6° (FWHM, full-width half maximum) (also called the small FoV), 6° × 6° (large FoV), and the blind FoV used to estimate the particle-induced instrumental background (see ref. [55] or The Insight-HXMT Data Reduction Guide for details). Since its launch, Insight-HXMT went through a series of performance verification tests by observing the blank sky, standard sources, and sources of interest. These tests showed that the entire satellite works smoothly and healthily, and have allowed for the calibration and estimation of the instruments background. The systematic errors of LE are <1% at 1–7 keV except for the Si K-edge at 1.839 keV. The response model of the energy band 7–10 keV of LE is well-calibrated based on more information of background lines of the detectors; For ME and HE, more accurate background models with HXMTDAS (v2.02, which will be public soon) are utilized and the effective area is updated. The systematic errors for ME 20–30 keV are relatively smaller. Fluorescence lines due to the photoelectric effect of electrons in Silver K-shell are detected by the Si-PIN detectors of ME, which dominates the spectrum over 21–24 keV. Therefore, the spectrum over 21–24 keV is ignored; The systematic errors of HE, compared to the model of the Crab nebular, are <2% at 28–120 keV and 2–10% above 120 keV. According to these calibration results, the energy bands for the spectral fits in this work are 2–10 keV for LE, 10–30 keV for ME, and 28–200 keV for HE, with a systematic error of 1.5% for LE/ME/HE.

We use the Insight-HXMT Data Analysis software (HXMTDAS) v2.0 to analyze all the data, filtering the data with the following criteria: (1) pointing offset angle < 0.1°; (2) pointing direction above Earth > 10°; (3) geomagnetic cut-off rigidity value > 8; (4) time since SAA passage > 300 s and time to next SAA passage > 300 s; (5) for LE observations, pointing direction above bright Earth > 30°. We only select events that belong to the small FoV for LE and ME observations, and for HE we use the events from both small and large FoV. To improve the signal-to-noise ratio of HE observation, we combine 17 spectra obtained from 15 small FoV and 2 large FoV together using addascaspec script in HEAsoft v6.8. The corresponding background spectra and response files are also combined together.

**Qualitative spectral analysis.** The spectra are analyzed using XSPEC version 12.10.0[56]. In Fig. 2, we plot the ratio of the unfolded Insight-HXMT spectrum of MAXI J1820 + 070 to the best-fitting cutoff powerlaw (cutoffpl in XSPEC) over 2–200 keV, for 2018-03-23 (MJD 58201), 2018-4-25 (MJD 58233), 2018-05-31 (MJD 58269), corresponding to the early, middle and late times of the decay, respectively. There is an increase in flux at low energies (~2–3 keV) relative to the powerlaw continuum. This ratio plot clearly reveals a broad Fe Kα emission line and a narrow component at around 6.4 keV. The broad component of the iron line visually appears stable throughout the decay, while the relative strength of the narrow core reduces with time. This is consistent with what is observed in refs. [28,33]. The hump at 20–100 keV is clearly seen in the ratio plot. The broad iron Kα emission line and the hump indicate the presence of relativistic reflection, as expected from an accretion disc extending close to a BH[31]. It is found that the amplitude of this hump drops along with the decay, while the broad iron line keeps fairly constant. This suggests that, the distant reflection component, which accounts for the narrow Fe K line and contributes to parts of the observed Compton hump, decreases in the normalization along with the decay. We will discuss this further in Supplementary Fig. 5.

**Configuration of the spectral fitting model.** In this work, we applied the relativistic model relxillCp which accounts for the reflection with the broad/high-ionization iron line and the non-relativistic model xillverCp which accounts for the reflection with the narrow/low-ionization iron line, to fit Insight-HXMT spectra. For both models, we use the same nthcomp parameters as the incident spectrum

inputs, and the seed photon temperature is fixed at[32] $kT_{bb} = 0.05$ keV. The inner radius, which is derived by fitting the NICER spectra with the relativistic reflection model, is estimated to be <2 $R_g$ under some assumptions[28]. It was also suggested that there is little or no evolution in the truncation radius of the inner disk during the luminous hard state, given that the thermal reverberation lags remain throughout all epochs and that the spectral shape of the broad Fe line component is constant over time. In ref. [33], the inner radius is fitted to be around 5 $R_g$ over the decay. If assuming the BH spin $a = 0.998$, this suggests that the inner disk may be truncated at small radius. Instead, if the accretion disk extends to the ISCO, i.e., $R_{in} = R_{ISCO} \simeq 5R_g$, where $R_{ISCO}$ is the ISCO, this may imply that the BH spin is low. Also, fits to the spectra of NICER in the soft state suggests a low spin[52]. Indeed, fixing $R_{in}$, spin, and inclination at different values, will result in different values for other free parameters, but the time-dependent trends in the fitted parameters will be preserved. Since in this work we are mainly concerned with the evolution of the corona, we assume the BH spin $a = 0.998$, and the inner radius is assumed to be $R_{in} = R_{ISCO}$. The effect of the low spin on the evolution of the corona will be discussed below. The outer radius of the reflection disk is fixed at the upper limit of their table model $R_{out} = 1000R_g$.

The strong absorption dips were detected by NICER in the outburst of MAXI J1820 + 070, suggesting that this source is a high-inclination system[24,37]. The observed sharp increase in the Hα emission-line EW and the absence of X-ray eclipses in MAXI J1820+070 in ref. [57] indicated the inclination angle to be 69° < i < 77°. The measurement of the radio parallax indicates that this source is located at a distance of 2.96 ± 0.33 kpc away from us. Together with the measured proper motions of the approaching and receding jet ejecta in radio, the inclination angle between the jet and the line of sight is estimated to be 63° (see ref. [21]). In this work, we take the inclination angle as 63°. The three free parameters of the relativistic reflection model relxillCp are the emissivity profile $q_1$, the reflection fraction $R_f$, the incident photon index Γ, the Fe abundance relative to the solar value $A_{Fe}$, the ionization parameter $log\xi$, and the electron temperature $kT_e$. The emissivity profile is parameterised in terms of the index $q_1$, $q_2$, and the broken radius $R_{br}$. For simplicity, the index of the outer disk region with $q_2 = 3.0$. The index of the inner disk region $q_1$ and the broken radius $R_{br}$ are free parameters. As for the non-relativistic reflection model, xillverCp, the spectral fits are insensitive to the ionization parameter, which is evaluated with F-test in XSPEC. Therefore, the ionization parameter is fixed at a low value with $log\xi = 1.0$, while the only free parameter is the normalization.

**Spectral fitting method**. The statistical analysis of all the fits are achieved by implementing a MCMC algorithm. More specifically, we used the MCMC algorithm implemented in XSPEC to create a chain of parameter values whose density gives the probability distribution for that parameter. For each individual spectrum fitting, two chains of 250,000 steps are run with 20 walkers. The first $10^4$ steps of the running are discarded as burn-in phase. Convergence between these two chains is assessed by Gelman-Rubin diagnostic[58], which requires $|R - 1| < 0.01$. The analysis of the probability distributions derived from the MCMC chains is implemented using the corner package[59], to estimate the mean values and errors of model parameters.

Supplementary Fig. 11 shows an illustration of one and two-dimensional projections of the posterior probability distributions derived from the MCMC analysis for the parameters in relxillCp, i.e., the emissivity profile $q_1$, the photon index Γ, the ionization parameter $log\xi$, the abundance $A_{Fe}$ with respect to the solar value, the electron temperature $kT_e$, the reflection fraction $R_f$, the normalization, the constant for ME and the constant for HE. The contours in the two-dimensional projections for each set of two parameters correspond to 1-, 2-, and 3-$\sigma$ confidence interval. The vertical lines in the one-dimensional projections correspond to the lower, median, and upper values of each parameter. The MCMC analysis shown in this figure is produced using the corner package[59]. This illustration corresponds to the spectral fitting (see Supplementary Fig. 1) of MJD 58204 (ObsID = P0114661006).

## Data availability
All Insight-HXMT data used in this work are publicly available and can be downloaded from the official website of Insight-HXMT: http://hxmtweb.ihep.ac.cn/.

## Code availability
The data reduction is done by the use of the software (HXMTDAS) v2.0 which is available at the Insight-HXMT website (http://en.hxmt.cn/analysis.jhtml). The addascaspec script in HEAsoft v6.8 is available: https://heasarc.gsfc.nasa.gov/docs/suzaku/analysis/addascaspec67.html. The model fitting of spectra and lag-energy spectra was completed with XSPEC[56], which is available at the HEASARC website (https://heasarc.gsfc.nasa.gov/xanadu/xspec/). The spectral models diskbb[34] and tbabs[35] are used. The reflection model relxill (v1.4.0)[32] used to fit the spectrum data is available: http://www.sternwarte.uni-erlangen.de/~dauser/research/relxill/.

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

## Acknowledgements

BY thanks J. García and T. Dauser for helping with the reflection model relxill, and thanks Z. Yan, X.G. Zheng, and S.X. Tian for the discussion of the spectral fitting. This work is supported by the National Program on Key Research and Development Project (Grants No. 2016YFA0400803) and the NSFC (11903024, U1931203, 11622326, U1838103, U1838201, and U1838202). This work made use of the data from the Insight-HXMT mission, a project funded by the China National Space Administration (CNSA) and the Chinese Academy of Sciences (CAS).

## Author contributions

B.Y., Y.L.T., C.Z.L., W.W., S.N.Z., S.Z., and C.L. contributed to spectral analysis and interpretation of results. M.Y.G. contributed to the calibration of Insight-HXMT data. B.F.L., W.M.Y., Z.G.D., J.F.L., E.L.Q., C.C.J., Z.L., and B.C. contributed to interpretation of results. B.Y., W.W., and S.N.Z. wrote the paper. T.P.L. is the previous PI (2000-2015) and S.N.Z. is the current PI of Insight-HXMT. Q.W.W., Q.C.B., C.C., X.L.C., Z.C., G.C., L.C., T.X.C., Y.B.C., Y.C., Y.P.C., W.C., W.W.C., J.K.D., Y.W.D., Y.Y.D., M.X.F., G.H.G., H.G., M.G., Y.D.G., J.G., C.C.G., D.W.H., Y.H., J.H., S.M.J., L.H.J., W.C.J., J.J., Y.J.J., L.D.K., B.L., C.K.L., G.L., M.S.L., T.P.L., W.L., X.L., X.B.L., X.F.L., Y.G.L., Z.W.L., X.H.L., J.Y.L., C.Z.L., G.Q.L., H.W.L., X.J.L., Y.N.L., B.L., F.J.L., X.F.L., Q.L., T.L., X.M., B.M., Y.N., J.Y.N., G.O., J.L.Q., N.S., R.C.S., L.M.S., X.Y.S., L.S., Y.T., L.T., C.W., G.F.W., J.W., L.J.W., W.S.W., Y.S.W., X.Y.W., B.Y.W., B.W., M.W., G.C.X., S.X., S.L.X., Y.P.X., J.W.Y., S.Y., Y.J.Y., Q.B.Y., Q.Q.Y., Y.Y., A.M.Z., C.M.Z., F.Z., H.M.Z., J.Z., T.Z., W.C.Z., W.Z., W.Z.Z., Y.Z., Y.F.Z., Y.J.Z., Y.Z., Z.Z., Z.L.Z., H.S.Z., X.F.Z., S.J.Z., D.K.Z., J.F.Z., Y.X.Z., and Y.Z. contributed to the development and scientific operation of Insight-HXMT.

## Competing interests

The authors declare no competing interests.

## Additional information

[1]School of Physics and Technology, Wuhan University, Wuhan, PR China. [2]Astronomical Center, Wuhan University, Wuhan, PR China. [3]Key Laboratory of Particle Astrophysics, Institute of High Energy Physics, Chinese Academy of Sciences, Beijing, PR China. [4]University of Chinese Academy of Sciences, Chinese Academy of Sciences, Beijing, PR China. [5]Key Laboratory of Space Astronomy and Technology, Chinese Academy of Sciences, Beijing, China. [6]School of Astronomy and Space Sciences, University of Chinese Academy of Sciences, Beijing, PR China. [7]School of Astronomy and Space Science, Nanjing University, Nanjing, PR China. [8]Key Laboratory of Optical Astronomy, National Astronomical

Observatories, Chinese Academy of Sciences, Beijing 100101, China. [9]WHU-NAOC Joint Center for Astronomy, Wuhan University, Wuhan, Hubei, China. [10]Center for Theoretical Physics, Polish Academy of Sciences, Warsaw, Poland. [11]School of Physics, Huazhong University of Science and Technology, Wuhan, China. [12]Tuebingen University, Tübingen, Germany. [13]Department of Astronomy, Beijing Normal University, Beijing, PR China. [14]Department of Physics, Tsinghua University, Beijing, PR China. [15]Department of Astronomy, Tsinghua University, Beijing, PR China. [16]Department of Engineering Physics, Tsinghua University, Beijing, PR China. [17]Computing Division, Institute of High Energy Physics, Chinese Academy of Sciences, Beijing, PR China. [18]School of Physics and Optoelectronics, Xiangtan University, Xiangtan, Hunan, China. [19]College of Physics, Jilin University, Changchun City, China. ✉email: wangwei2017@whu.edu.cn; zhangsn@ihep.ac.cn

