## [Peer Review File · Nature Communications]

Editorial Note: Parts of this peer review file have been redacted as indicated to maintain confidentiality.

REVIEWER COMMENTS

Reviewer #1 (Remarks to the Author):

I think the authors have addressed the remaining comments satisfactorily and this paper is worth publishing in Nature Communications.

Reviewer #2 (Remarks to the Author):

I have read the paper entitled 'The jet-like corona in a black hole X-ray binary observed by Insight-HXMT' and offer the following review:

This paper describes the evolution of the spectrum of the X-ray binary MAXI J1820+070 during the hard state. There are several other works describing this phase of this outburst (which are properly cited); the new features of this work are the detailed spectral analysis of the HXMT data and the interpretation of the changes as being due to outflowing material in the corona.

If confirmed, this is a very interesting interpretation: the corona being associated with the jet is a common suggestion but observational evidence for the outflowing corona which could be associated with this is rare.

However, there are some aspects of the spectral fitting which I feel need to be addressed before the paper can be accepted:

1. The paper uses models outside of their strict domain of applicability, without sufficient justification that this is appropriate:

The authors fit the spectra with `relxillpCp` (a stationary corona) for their main results, using the reflection fraction implied by the fitted height as a proxy for the velocity of a corona with a different true height, given by trends in timing properties.

However, the true reflection spectrum for a low, outflowing source will not in general match that for a high stationary source, since the emissivity profile as well as the reflection fraction differs. See for example Wilkins et al. 2012 (particularly Sec 3.5).

Since they use of `relxillpionCp` (a moving corona) to produce simulated spectra to confirm trends, why not perform all fits with `relxillpionCp`?

An alternative would be to fit for a parameterised emissivity profile in reflection models (e.g. the standard `relxill`) and testing whether an emissivity profile matching outflowing coronae (such as those from Wilkins et al. 2012) fits well (Wilkins et al. 2012 argue against coronae often being outflowing, since such profiles are typically not seen). The emissivity profile could also be fitted directly, as in Wilkins et al. (2011).

When comparing the fit to `relxillpionCp`, allowing only the height to be free is likely to be overly restrictive: since we do not know the true values of other parameters for the real data, this should be reflected in the simulated fits, as other parameters may change to offset the change due to source velocity, which could lead to a different behaviour of the height.

Some more minor related points:

- The conversion from height to reflection fraction is not really self-consistent, since the central claim of this work is that this actually represents a moving source at a different height to the measured height.

- For the standard `relxillp(Cp)`, reflection fractions lower than 1 for high coronal heights are significantly due to the finite outer radius of the disc; this should be discussed.

2. There are still some significant differences between HMXT and NuSTAR:

Line 72-3:

'Moreover, the best-fitting height of the X-ray point source decreases to about 7 R_g in the rise phase and increases from about 7 R_g to 500 R_g in the decay phase.'

This parameter is not actually shown in the figures. I feel it would be clearer to show it alongside the reflection fraction which is inferred from it.

Additionally, such an increase of the coronal height is not seen in the NuSTAR data of the same period, either in the analysis by Buisson et al. (2019) or the authors' own re-analysis.

Why does a change from 7-500 r_g in HMXT correspond to a 15-50 r_g change in NuSTAR?

3. Several parameters are frozen where they need not be, and some of the justifications for the values chosen cite previous work incorrectly:

Line 266: 'Additionally, in ref34 [Buisson et al. 2019] , the spin is assumed to have the extreme value $a = 0.998$,': this is not quite correct. The fits are performed with $a=0.998$ but R_{in} is not frozen to R_{ISCO} , so the fitted value, $\sim 5r_g$, could be due to disc truncation or to the spin actually being lower: due to the constancy of R_{in} , it is argued that a low spin (so $r_{ISCO} \sim 5r_g$) is more likely.

[redacted]

Also, fits to the soft state by Fabian et al. (2020) give a low spin.

I recommend that the authors check by performing some fits with a low spin and/or perform some fits to the HMXT data to directly determine the spin value preferred by this dataset.

Similarly, the inclination is fixed to the value from the jet/optical measurements. Since the inner disc/jet/orbital plane do not necessarily align, I recommend that the authors check (by fitting with the inclination free) that the HMXT data prefers a high inclination, and that they cannot get similar/better fits for a low inclination (as has been found in Buisson et al. 2019 and Fabian et al. 2020).

Some minor points:

Line 84:

'However, the height estimated by the timing analysis of NICER data covering the same observation period, decreases with time, in contrast'

Kara et al. (2019) only cover the first part of the decay. However, from analysing later NICER data, I find that there is no large reversal in the trend, so the stated contrast remains.

Line 250-1:

'This suggests that, the distant reflection component, which accounts for the narrow Fe K line and the Compton hump'

The reflecting material which produces the broad part of the iron line should also contribute to the Compton hump.

line 275-6: the mass function does not give the inclination; various other arguments in ref 41 give the stated range.

Lines 288-311 repeat lines 254-288.

Line 421: ite -> its; th eend -> the end

Line 432: 'In ref34 , the inner radius is one of the free parameters'

For clarity, this is the inner radius from the shape of the reflection spectrum, not from the disk parameters.

Line 445 contrasts -> contracts

NuSTAR table: Why are the two norms given as relating to FPMA and FPMB? If this is correct, they should not differ by so much (maximum 5%). Are they actually the norms for the upper and lower components?

Incidentally, the differences in the diskbb components between FPMs have now been identified as due to a thermal blanket tear in FPMA (Madsen et al. 2020).

Additional comments of Reviewer#2 during consultation:

My comments on the responses to the comments of the previous reviewer 2:

1. I am happy with this response
2. See my comment 3
3. I am happy with this response (but note that reviewer 1 may have other insights, given their similar concern)
4. I am happy with the changes in the discussion of the trends in the NuSTAR data, but see also my comment 3.

We appreciate the two Reviewers (#1,#2) for helpful report. The revisions and response to the referee's reports are listed below. All the revisions in the text of this revised version are in boldface.

Reviewer #1 (Remarks to the Author):

I think the authors have addressed the remaining comments satisfactorily and this paper is worth publishing in Nature Communications.

Answer:

We are very grateful for the suggestions and comments by the reviewer, which greatly improve our manuscript.

Reviewer #2 (Remarks to the Author):

I have read the paper entitled 'The jet-like corona in a black hole X-ray binary observed by Insight-HXMT' and offer the following review:

This paper describes the evolution of the spectrum of the X-ray binary MAXI J1820+070 during the hard state. There are several other works describing this phase of this outburst (which are properly cited); the new features of this work are the detailed spectral analysis of the HXMT data and the interpretation of the changes as being due to outflowing material in the corona.

If confirmed, this is a very interesting interpretation: the corona being associated with the jet is a common suggestion but observational evidence for the outflowing corona which could be associated with this is rare.

However, there are some aspects of the spectral fitting which I feel need to be addressed before the paper can be accepted:

1.

The paper uses models outside of their strict domain of applicability, without sufficient justification that this is appropriate: The authors fit the spectra with `relxilllpCp` (a stationary corona) for their main results, using the reflection fraction implied by the fitted height as a proxy for the velocity of a corona with a different true height, given by trends in timing properties. However, the true reflection spectrum for a low, outflowing source will not in general match that for a high stationary source, since the emissivity profile as well as the reflection fraction differs. See for example Wilkins et al. 2012 (particularly Sec 3.5). Since they use of `relxillpionCp` (a moving corona) to produce simulated spectra to confirm trends, why not perform all fits with `relxillpionCp`? An alternative would be to fit for a parameterised emissivity profile in reflection models (e.g. the standard `relxill`) and testing whether an emissivity profile matching outflowing coronae (such

as those from Wilkins et al. 2012) fits well (Wilkins et al. 2012 argue against coronae often being outflowing, since such profiles are typically not seen). The emissivity profile could also be fitted directly, as in Wilkins et al. (2011). When comparing the fit to `relxillpionCp`, allowing only the height to be free is likely to be overly restrictive: since we do not know the true values of other parameters for the real data, this should be reflected in the simulated fits, as other parameters may change to offset the change due to source velocity, which could lead to a different behaviour of the height.

Answer:

We fully agree with the referee. Wilkins et al. (2012) systematically studied the emissivity profile of the accretion disk, in the cases of different size/geometries and the relativistically outflowing motion of the X-ray primary source. It was clearly shown that, the emissivity profile for a low/outflowing source and a high/static source are different with respect to each other, and therefore the reflection spectra will differ correspondingly.

We tried to use `relxillpCp` (a stationary corona) and `relxillpionCp` (a moving corona) to perform the spectral fits:

First, we take the spectrum of MJD =58204 (obsID = P0114661006) as an example, which roughly corresponds to the peak of the outburst. The best-fitted height is about $H = 17.15 R_g$, and the corona is stationary with the fitted velocity $v=0$ (case 1). Thus, the reflection fraction $R_f = 1.2$.

Secondly, we take the spectrum of MJD =58271 (obsID = P0114661061) as another example, which corresponds to the later epoch of this decay, when the corona was believed to be contracting closer to the black hole (Kara2019 and the information from the referee). We found that, case (2.1)–a low corona ($H \sim 17 R_g$) with the fast velocity ($v \sim 0.5c$, the reflection fraction $R_f = 0.6$), or case (2.2)–a relatively higher corona (e.g. $H \sim 45 R_g$) with the relatively slower velocity ($v \sim 0.4c$, the reflection fraction $R_f = 0.55$), can both provide the equivalently good fits to the data (see Fig. 1 below).

Finally, we refit the spectrum of MJD =58271 again, but fix the velocity $v=0$ (corresponding to the `relxillpCp` model), then the fitted height $H \sim 3 R_g$ (case3: $v=0$, the reflection fraction $R_f = 3.3$) and this fitting is somewhat worse with $\Delta\chi^2 = 8$.

By comparing case (3) with case (2.1, 2.2), it can be seen that the corresponding reflection fraction is different. This means that the conversion from height to reflection fraction is indeed not really self-consistent between a stationary corona and a moving corona (as the referee mentioned above), since the central claim of this work is that this actually represents a moving source at a different height to the measured height. Therefore, using the height from the stationary corona (relxilllpCp) model to represent the evolution of the reflection fraction is not appropriate. Then, we consider relxillpionCp and relxillCp

By comparing case (2.1) with case (2.2), in the case of a moving corona (relxillpionCp), it can be seen in Fig. 1 that, the height and the outflowing velocity cannot be uniquely determined from the spectral fits. But, it is interesting that, the corresponding reflection fractions are roughly the same $R_f \sim 0.6$ and the velocity is moderate $v \sim 0.5c$, at the end of this decay. Furthermore, by comparing case (1) with case (2.2 an 2.2), it can be seen that, at the peak of this decay, the reflection fraction is large with $R_f = 1.2$ and the corona is fitted to be stationary. These comparisons suggest that, by the use of the relxillionCp,, the reflection fraction decreases while the outflowing velocity increases, during the decay.

Although we cannot get the unique height and velocity of the corona with HXMT spectral fits using relxilllpionCp, it is still interesting to infer the evolution of the corona outflowing for a given corona height, as discussed above. Therefore, we follow the referee suggestion to use the relxilllpionCp to perform the spectral fits but fixing the height of the corona (e.g., $H = 7 R_g$) for simplicity. It turns out that the corona is outflowing faster with the decay (see Fig. 16).

Given the reflection fraction and emissivity profile computationally affect the reflection spectrum, we could directly fit them by the use of the standard reflection model (relxillCp). Therefore, we follow the referee suggestion to use the standard relxill to fit for a parameterized emissivity profile and the reflection fraction, and testing whether the derived values of these parameters support the claim of the outflowing corone, considering the predictions by Wilkins et al. (2012). In the relxill model, the emissivity profile is assumed to be broken powerlaw of two indexes q_1 and q_2 with the broken radius R_{br} . In our spectral fits, for simplicity, we assume the index $q_2 = 3$ for the region where $R > R_{br}$. It turns out, the emissivity profile is roughly flattened with the index $q_1 < 1$ within the broken radius $R_{br} \sim 20-60 R_{isco}$ (where $R_{isco} \sim 1.23 R_g$ for $a = 0.998$). Providing the predictions by Wilkins et al. (2012; e.g., their Fig 10, 12), the flattened emissivity profile suggest that the X-ray corona source might be spatially extended with the size of tens of R_g in radius. According to the theoretical computation (Wilkins et al. 2012), the emissivity profile for the point source is predicted to be much steep with the index $q \sim 6-8$ in the inner region (e.g, $R < 3 R_g$) before the profile flattens off (where $R < R_{br}$). Such a twice-broken powerlaw profile was observed in the narrow-line Seyfert 1 galaxy,

1H 0707-495 by fitting the relativistically broadened emission lines from the disk (Wilkins & Fabian 2011). The derived small index (< 1) in the inner disk region from our spectral fits may be due to (1) the relxill use a single broken-powerlaw, so the index q_1 is the emissivity-weighted value for the region $R < R_{br}$; (2) the index q_1 and the inner radius are somewhat degenerate with each other, in the sense that, on one hand, if the emissivity profile falls steep with the index $q_1 \sim 6-8$, then in order to correctly reproduce the flux levels emitted from the innermost regions, the disk must be truncated at a larger radius; On the other hand, if the emissivity profile is flat with the small index $q_1 < 1$, then the disk can extend to the innermost regions with small truncation radius, to match the flux levels (Wilkins et al. 2012). In our spectral fits, the inner radius is assumed to be small with $R_{in} = R_{isco}$, for simplicity. If we assume $R_{in} = 20 R_g$ instead and refit the spectra, the emissivity profiles then are required to be steep with the index $q_1 \sim 6-8$.

Meanwhile, the reflection fraction is smaller than unity, which requires the spatially extended corona to be relativistically outflowing to overcome the increasingly important light-bending effect, since the corona is suggested to be collapsing by NICER lag-observations (Kara 2019).

In this revision, the spectral fit results with the standard relxill model are taken as the main results (Line 50-51 on Page 5; Line 72-73 on Page 6; Line 257-259 on Page 13; Line 283-291 on Page 14; Line 311-334 on Page 16).

Some more minor related points:

- The conversion from height to reflection fraction is not really self-consistent, since the central claim of this work is that this actually represents a moving source at a different height to the measured height.

Answer:

The referee is right. As discussed above, taking the spectrum of MJD=58271 (obsID = P0114661061) as an example, by comparing case (3) with case (2.1, 2.2), we show that the corresponding reflection fraction is different. This means that the conversion from height to reflection fraction in `relxillpCp` is indeed not really self-consistent (as the referee mentioned above), since the central claim of this work is that this actually represents a moving source at a different height to the measured height. Therefore, using the height from the stationary corona (`relxillpCp`) model to represent the evolution of the reflection fraction is not appropriate.

Therefore, in this revision, following the referee suggestion above, we directly fit the reflection fraction from the data with the standard `relxill` model. We discuss this at line 494–505 of Page 23.

– For the standard `relxillp(Cp)`, reflection fractions lower than 1 for high coronal heights are significantly due to the finite outer radius of the disc; this should be discussed.

Answer:

Yes. Indeed, in the `relxillp(Cp)` for the point-like lamppost, the disk outer radius cannot exceed $1000 R_g$. Therefore, when the corona is high, e.g., $H > 100$, a fraction of photons will not hit the disk, which results in the reflection fraction being lower than unity. This should be distinguished from the case of the corona being relativistically outflowing at low height ($H < 100$). In the latter case, the reflection fraction will also be small with $R_f < 1$, but now it is due to the beaming effect of the relativistic outflowing motion (see Fig 4).

We discuss this at line 487–493 of Page 23.

2.

There are still some significant differences between HMXT and NuSTAR:

Line 72–3:

‘Moreover, the best-fitting height of the X-ray point source decreases to about $7 R_g$ in the rise phase and increases from about $7 R_g$ to $500 R_g$ in the decay phase.’

This parameter is not actually shown in the figures. I feel it would be clearer to show it alongside the reflection fraction which is inferred from it.

Additionally, such an increase of the coronal height is not seen in the NuSTAR data of the same period, either in the analysis by Buisson et al. (2019) or the authors’

own re-analysis.

Why does a change from 7-500r_g in HXMT correspond to a 15-50r_g change in NuSTAR?

Answer:

Thank the referee for raising up this concern, which leads us to recheck the spectral fits with the lamppost model. By comparing the results of us with Buisson (2019), we discovered the differences of the model configurations and assumptions:

In our previous lamppost fits, we fixed the height and ionization parameter of the upper lamppost at the constant values, i.e, $H_2 = 500$ and $\log \xi_2 = 1.0$, which is inappropriate, since the spectral fits are sensitive to these parameters; Instead, in Buisson (2019), these two parameters are allowed to be free. Now, we make the same setting to these two parameters as Buisson (2019).

In Buisson (2019), the heights of both the lower and upper lamppost are fitted within 100 R_g (This can be inferred from their Fig. 7 and the errors of the height in Table 7). Since the intrinsic range of the height is not known in advance, now we release this limit of the height.

After making the above modifications of the model configuration, we refit HXMT and NuSTAR. It turns out that, for both low and high inclination, NuSTAR and HXMT observed the same decrease in the heights of the upper lamppost (H_2). Therefore, using the lamppost model, the evolution of the corona from HXMT data is now consistent with the one from NuSTAR data.

In this figure, H_1 and H_2 are the heights of the lower and upper lampposts. It has been shown in Buisson(2019) that, for low inclination, the height of the upper lamppost decreases during the decay phase. Here we fit the heights for the case of high inclination angle ($i = 63$), given the radio/optical measurements. It can be seen that the height of the upper lamppost still decreases during the decay.

Coming back to the referee's main concern of this work, our further studies above confirm that the decrease of the height alone cannot explain the observed reflection evolution, which also requires the outflowing corona reported in this work. As the referee also mentioned above, the conversion from height to reflection fraction is not really self-consistent, since the central claim of this work is that this actually represents a moving source at a different height to the measured height (see Fig 1 above in this response letter).

We therefore follow the referee's suggestion to fit the spectrum with the standard reflection model. The fitted reflection fraction for both HXMT and NuSTAR clearly decreases during the decay (Fig 3 and 19). In Fig 19, each point corresponds to the spectral fits for obsID in Table 1 of Buisson(2019).

We discuss this at Line 485-487, and 494-524 on Page 23, 24.

3.

Several parameters are frozen where they need not be, and some of the justifications for the values chosen cite previous work incorrectly:

Line 266: 'Additionally, in ref34 [Buisson et al. 2019] , the spin is assumed to have the extreme value $a = 0.998$,': this is not quite correct. The fits are performed with $a=0.998$ but R_{in} is not frozen to R_{ISCO} , so the fitted value, $\sim 5r_g$, could be due to disc truncation or to the spin actually being lower: due to the constancy of R_{in} , it is argued that a low spin (so $r_{ISCO} \sim 5r_g$) is more likely.

[Redacted]

Also, fits to the soft state by Fabian et al. (2020) give a low spin. I recommend that the authors check by performing some fits with a low spin and/or perform some fits to the HXMT data to directly determine the spin value preferred by this dataset.

Answer:

Following the referee's suggestion, we refit six observations from the data sets, which covers the decay of this outburst. We use the same model as in the main results, but fixing the spin $a=0.5$. It turns that the main results do not change (see Fig 15), i.e., the reflection fraction is also small than unity, and decreases during the decay, which requires the relativistic motion outwards. We did not fit the data to determine the spin directly, since our results are not sensitive to this.

We discuss this at Line 444-451 on Page 21.

Similarly, the inclination is fixed to the value from the jet/optical measurements. Since the inner disc/jet/orbital plane do not necessarily align, I recommend that

the authors check (by fitting with the inclination free) that the HXMT data prefers a high inclination, and that they cannot get similar/better fits for a low

inclination (as has been found in Buisson et al. 2019 and Fabian et al. 2020).

Answer:

Following the referee's suggestion, we refit the same six observations as above, which covers the decay of this outburst, but allowing the inclination to be free. This gives better fits., and the fits prefer to the large values of the inclination angle (see Fig 16). These estimates are in agreement with the jet/optical measurements, although it cannot be quite certain with the intrinsic inclination angle (Fabian 2020). Nevertheless our results are not sensitive to the assumed inclination and thus our conclusion is unchanged.

We discuss this at Line 452-458 on Page 22.

Some minor points:

Line 84:

'However, the height estimated by the timing analysis of NICER data covering the same observation period, decreases with time, in contrast'

Kara et al. (2019) only cover the first part of the decay. However, from analysing later NICER data, I find that there is no large reversal in the trend, so the stated contrast remains.

Answer:

Thank the referee for reminding us of the updated analysis of later NICER data, and we are happy to know that the trend remains during the later decay. In the revision, we modify the statement to clearly state that Kara et al. (2019) only cover the first part of the decay, at Line 84-86 on Page 6.

Line 250-1:

'This suggests that, the distant reflection component, which accounts for the narrow Fe K line and the Compton hump'

The reflecting material which produces the broad part of the iron line should also contribute to the Compton hump.

Answer:

Thank the referee remind of this inaccurate statement. In this revision, we modify it as follows:

"This suggests that, the distant reflection component, which accounts for the narrow Fe K line and contributes to parts of the observed Compton hump" (Please see Line 253-255 on Page 13)

line 275-6: the mass function does not give the inclination; various other arguments in ref 41 give the stated range.

Answer:

Thank the referee for reminding us of this inaccurate statement. In this revision, we modify it as follows:

“The observed sharp increase in the Ha emission line equivalent width and the absence of X-ray eclipses in MAXI J1820+070 in ref \cite{Torres2019} indicated the inclination angle to be $69 < i < 77$ ” (Please see Line 276-2s79 on Page 14)

Lines 288-311 repeat lines 254-288.

Answer:

Thanks a lot ! We delete the repeated lines 288-311

Line 421: ite -> its; th eend -> the end

Answer:

Done

Line 432: 'In ref 34 , the inner radius is one of the free parameters'

For clarity, this is the inner radius from the shape of the reflection spectrum, not from the disk parameters.

Answer:

Done. Please see Line 423-424 on Page 21.

Line 445 contrasts -> contracts

Answer:

Done

NuSTAR table: Why are the two norms given as relating to FPMA and FPMB? If this is correct, they should not differ by so much (maximum 5%). Are they actually the norms for the upper and lower components?

Incidentally, the differences in the diskbb components between FPMs have now been identified as due to a thermal blanket tear in FPMA (Madsen et al. 2020).

Answer:

Thank the referee for sharing this information of Madsen et al. (2020) with us. We cite this paper in the revision to clarify the treatment of the differences in the diskbb components between FPMs (Line 517–518 on Page 24).

The two norms in Table 2 represent the norms for the upper and lower components. We know that in Buisson (2019), the normalizations of the two lampposts are tied together to fit data. It was assumed in that work that the X-ray source is vertically extended. The total model spectrum comes from the radiation of the lower and upper lamppost with the implicit assumption that the two components have the same radiation luminosity. However, the radiation of these two components is not necessarily the same. And, we have checked that, tie or untie these two normalizations in the lamppost model, do not change the evolution of the heights.

REVIEWERS' COMMENTS

Reviewer #2 (Remarks to the Author):

The authors have addressed my previous concerns and I am happy for this paper to be published.